# Breeding of Vegetable Cowpea for Nutrition and Climate Resilience in Sub-Saharan Africa: Progress, Opportunities, and Challenges

**DOI:** 10.3390/plants11121583

**Published:** 2022-06-15

**Authors:** Tesfaye Walle Mekonnen, Abe Shegro Gerrano, Ntombokulunga Wedy Mbuma, Maryke Tine Labuschagne

**Affiliations:** 1Department of Plant Sciences, University of the Free State, Bloemfontein 9301, South Africa; mbumanw@ufs.ac.za (N.W.M.); labuscm@ufs.ac.za (M.T.L.); 2Agricultural Research Council-Vegetable, Industrial and Medicinal Plants, Pretoria 0001, South Africa; agerrano@arc.agric.za; 3Food Security and Safety Focus Area, Faculty of Natural and Agricultural Sciences, North-West University, Mmabatho 2735, South Africa

**Keywords:** climate change, cowpea, food, gene pyramiding, nutrition security, speed breeding

## Abstract

Currently, the world population is increasing, and humanity is facing food and nutritional scarcity. Climate change and variability are a major threat to global food and nutritional security, reducing crop productivity in the tropical and subtropical regions of the globe. Cowpea has the potential to make a significant contribution to global food and nutritional security. In addition, it can be part of a sustainable food system, being a genetic resource for future crop improvement, contributing to resilience and improving agricultural sustainability under climate change conditions. In malnutrition prone regions of sub-Saharan Africa (SSA) countries, cowpea has become a strategic dryland legume crop for addressing food insecurity and malnutrition. Therefore, this review aims to assess the contribution of cowpea to SSA countries as a climate-resilient crop and the existing production challenges and perspectives. Cowpea leaves and immature pods are rich in diverse nutrients, with high levels of protein, vitamins, macro and micronutrients, minerals, fiber, and carbohydrates compared to its grain. In addition, cowpea is truly a multifunctional crop for maintaining good health and for reducing non-communicable human diseases. However, as a leafy vegetable, cowpea has not been researched and promoted sufficiently because it has not been promoted as a food security crop due to its low yield potential, susceptibility to biotic and abiotic stresses, quality assurance issues, policy regulation, and cultural beliefs (it is considered a livestock feed). The development of superior cowpea as a leafy vegetable can be approached in different ways, such as conventional breeding and gene stacking, speed breeding, mutation breeding, space breeding, demand-led breeding, a pan-omics approach, and local government policies. The successful breeding of cowpea genotypes that are high-yielding with a good nutritional value as well as having resistance to biotics and tolerant to abiotic stress could also be used to address food security and malnutrition-related challenges in sub-Saharan Africa.

## 1. Introduction

Cowpea [*Vigna unguiculata* (L.) Walp.] is a nutritious, underutilized vegetable legume crop that has the potential to alleviate protein-calorie malnutrition [1]. The crop can be grown under harsh conditions such as drought and sandy soils. Thus, its ability to tolerate climate change makes it an important legume crop for food and nutritional security in tropical and sub-tropical regions of the world, including in sub-Saharan Africa (SSA) [1,2,3,4]. Cowpea is known in its dry grain form as black-eyed pea, southern bean, China pea and marble pea, and in its green pod form as a yard-long bean, an asparagus bean, a body bean and a snake bean [5,6]. The hulls of cowpea are fed to livestock as a nutritious supplement to cereal fodder [7]. Cowpea contributes to the sustainable improvement of the environment [8] due to its biological nitrogen fixation ability and its positive effects on the soil [9], as it increases microbial diversity in the soil and plays a major role in resilience to current climate changes in the region and beyond. Due to the high protein content in its leaves, immature pods and grain, cowpea is a potential crop for reducing the deficiency in plant-based protein production in SSA [10]. Despite its role in the food system value chain, there is limited information documented for future sustainable production and improvement of this crop for nutritional diversity and on utilization and key production constraints that affect the performance of the crop in SSA.

## 2. Use of Cowpea as a Leafy Vegetable and Grain Crop

It has been reported that about 150 million children in the world under five suffered from stunting in 2020. Forty-five million of these children were wasted. Approximately 39 million children globally under five years of age are overweight [11]. Childhood obesity is also currently challenging and is affecting the economy and development of the world population [12]. It has been reported that one in four children in South Africa suffers from severe acute malnutrition (SAM), which remains a fundamental severe cause of child morbidity [13]. The South African Child Gauge report indicated that 44% of children under the age of five suffer from vitamin A deficiency, and 27% are stunted [13]. Meanwhile, 33.3% of women aged 15–49 years are anemic, while 31.3% of males (≥15 years) and 67.6% of females (≥15 years) are either overweight or obese [14]. Regarding food security, about 56% of the population in South Africa live in poverty and almost 28% live in extreme poverty [15]. Several nutrition-sensitive approaches are being introduced in response to malnutrition and the food insecurity burden [16].

Cowpea is an underutilized plant species that can contribute to food, nutrition, and health security in SSA. Cowpea is consumed in many forms; its young leaves, growing points, green pods, and green seeds are all used as vegetables in SSA [10]. Dry seeds are used in various food preparations that have been recommended for the possible alleviation of food and nutrition insecurity [10]. Cowpea leaves can be dried and used as a meat substitute in the rural areas of Africa. Cowpea flour provides extra protein and nutrition that enhances the growth and development of healthy children.

The fresh and young leaves of cowpea, which is ranked among the top four vegetables in 24 countries in Africa and seven in Asia, are suited for production in high rainfall agro-ecologies [17]. Hence, the use of cowpea as both a vegetable and grain crop can contribute to reduced malnutrition in the region.

Globally, cowpea production has been reported to have increased by 88% and yields have increased by 35% [18]. The total cowpea area harvested has also risen by 38% in the same production period [19]. This increase in area, production, and yield has been made possible by a similar trend in SSA, which dominates the world scene [20]. In the last three decades, cowpea’s area coverage and production potential has increased significantly (Figure 1) [20]. Sub-Saharan Africa dominates cowpea production, with a 96% (4.9 million tons) area share [19,21]. Total area, yield, and production in SSA have grown at the rate of about 4.3%, 1.5%, and 5.8%, respectively [19,20]. Generally, the area under cowpea has increased, with many developing countries contributing to the total production from the last decade onwards [19]. Over 95% of the global production is in Africa, especially in SSA, with Nigeria being the world’s largest producer and consumer, followed by the Niger Republic and Burkina Faso [22].

In sub-Saharan Africa, the major legume crops in the region are groundnuts, chickpea, pigeonpea, common bean, cowpea, and soybean. In SSA, the area coverage was about 36 million hectares (about 16.3% of global area), with production of about 27 million metric tons (around 6% of global production) at an average productivity of 0.89 metric tons (MT) per hectare (beans = 0.94, chickpea = 0.98, cowpea = 0.48, groundnut = 0.96, pigeon pea = 0.86, soybean = 1.01) [20,21]. The SSA region accounted for more than 99% of the cowpea area and 96% of production, with an average yield of about 0.52 MT per ha in 2020 [19,21]. Cowpea production in SSA is projected to grow from about 9.8 million MT in 2020 to nearly 12.3 million MT by 2030 [19,21].

## 3. Constraints to Cowpea Production

Climate change and variability trends affect crop yields both directly and indirectly [23]. Direct impacts include effects caused by a modification of physical characteristics such as low or high atmospheric temperature, soil fertility level, and water deficiency or erratic rainfall distribution in specific crop production systems [24]. Indirect effects are those that affect production through changes in other species such as pollinators, insect pests, diseases, and weeds. These indirect effects can play a significant role in the production and productivity of cowpea [24]. These limiting factors can broadly be termed abiotic and biotic stresses, resulting in climatic variations and ultimately reducing cowpea yield potential and its productivity [25,26,27].

### 3.1. Biotic Stresses

#### 3.1.1. Bacterial Diseases

Bacterial disease is the most economically important biotic stress in cowpea production in SSA countries. Among them, cowpea bacterial blight (CoBB) and bacterial pustules are some of the most severe bacterial infections of cowpea, causing severe damage [26,27]. CoBB symptoms start with small water-soaked spots on leaves, which enlarge to irregular brown necrotic lesions surrounded by yellow haloes, leading to premature leaf drop [26]. The pathogen also invades the cowpea stem, causing canker symptoms in susceptible plants [27]. It generally infects all growth stages of the cowpea plant: seedling, vegetative, flowering, and podding stages, and all plant parts, including leaves, pods, and seeds [28]. Frequent evaluation of cowpea genetic materials and resistant line development is the best option for achieving desirable resistance gene sources for use in bacterial resistance through breeding programs.

#### 3.1.2. Viral Diseases

Globally, cowpea is infected by more than 140 viruses, but only nine were reported as economically important [29]. Cowpea aphid-borne mosaic virus (CABMV), cowpea golden mosaic virus (CPGMV), Southern bean mosaic virus (SBMV), Sunhemp mosaic virus (SHMV), blackeye mosaic virus (BICMV), cucumber mosaic virus (CMV), cowpea mottle virus (CMV), cowpea yellow mosaic virus (CPMV), and cowpea mild mottle virus (CPMMV) are the most common seed-borne viral diseases [30]. Viral diseases can cause a yield loss of 10% to 100%, depending on the time and the severity of infection in cowpea [31]. The aphids suck the cowpea sap, which might affect the physiological processes and mineral element transportation in the plant system and consequently affect its concentration [10]. It is also a vector in the transmission of viruses [32]. For tackle the existing viral disease problem, intensive evaluation of cowpea genotypes with multiple viral infections under greenhouse and hotspot production areas are the best option to develop resistant varieties. In addition, using biotechnology that will result in the production and deregulation of virus-resistant cowpeas through coat-protein gene transfers should be intensified [29].

#### 3.1.3. Root-Knot Nematode

Root-knot nematode, *Meloidogyne incognita*, is a severe pest and a major constraint on cowpea production in most growing areas of the world, causing 80 to 100% yield losses [32,33,34,35,36,37,38,39,40,41,42,43,44,45,46,47,48]. *M. incognita* and *M. javanica* are the major species found on cowpea in most growing regions [33]. Damage symptoms of root-knot nematodes include patches of stunted and yellowed plants. Severe damage can lead to reduced numbers of leaves and buds [5].

#### 3.1.4. Parasitic Weeds

Parasitic weeds, *Striga gesnerioides* (Willd) Vatke ex Engl and *Alectra vogelii* (Bent), are serious threats to cowpea production in Africa [31,32]. Weeds reduce cowpea yield and quality by competing for light, space, water, soil nutrients, and carbon dioxide [33,34]. In total, yield losses caused by parasitic weed infestation alone in cowpea ranges from 73 to 100% [34,35,47]. These weeds may also reduce productivity by releasing allelopathic compounds into the environment [40] and by providing a conducive environment and serving as a vector for insect pests and viruses [41].

Completely removing these parasitic weeds in cowpea production is too difficult because the seeds can remain viable in the soil for up to 20 years [31,39]. Therefore, the best and most sustainable option to address this critical problem is to breed cowpea varieties which are resistant to these weeds, using multiple smart and agro-biotechnology techniques that could be deployed sustainably.

#### 3.1.5. Insect Pests

Cowpea is attacked and damaged by insect pests in all stages of growth [43]. Insects are the most challenging threat to cowpea production and productivity because they occur at pre-flowering, post-flowering, and storage [44]. Seed com, maggot, cutworm, aphids, and leafhopper occur at the pre-flowering (seedling) stage. Aphids, leaf miners, and thrips are also active insect pests at the flowering and post-flowering stage, and aphids, bean fly, bean pod borer, leaf miner, and thrips are the common insect pests at the reproductive stage (grain filling period) of cowpea [45]. Among the field pests of cowpea, aphid [*Aphis craccivora* (Koch)] is an important vegetative stage pest of cowpea in Africa but also occurs at other growth stages. Both nymphs and adults suck plant sap and cause serious damage from the seedling to the pod bearing stage [46]. Aphids cause damage through secretion of honeydew, which promotes the growth of sooty molds and other fungi on leaves, curling of leaves and delayed flowering, shriveling of pods and, as a result, reduced photosynthetic processes and rates, finally resulting in overall yield reduction [47]. It affects the crop by directly sucking its sap. Their feeding on cowpea causes cupping of the leaves, crinkling, defoliation, and stunted growth [48]. Another serious effect of aphids is the ability to transmit the aphid-borne mosaic virus. Affected plants show a green vein banding of the leaves [49]. Molecular and phenotypic screening of cowpea [35] identified cowpea genotypes with good resistance to aphids, which can be used as a source of resistance genes in breeding new aphid-resistant cultivars.

### 3.2. Abiotic Stress

#### 3.2.1. Drought

Cowpea is known to be drought-tolerant compared to other crop plants [50,51]. Among the abiotic factors, drought has been identified as a significant limitation, restricting cowpea production and productivity [50]. In the arid and semi-arid tropics, the productivity of cowpea could be hampered by erratic rainfall at the beginning and towards the end of the rainy season [52]. Drought leads to adverse influences on cowpea growth, development, and reproduction ability [53], limiting the crop’s yield and productivity [54]. This crop plant is a robust legume; nevertheless, drought always affects yield, especially during the reproductive and seed-filling period [55]. This causes a substantial reduction in grain yield and biomass production [51]. The influence of drought varies and depends on the intensity, developmental stage, and duration of stress and the adaptive strategy that the plant possesses to tolerate this stress [51,56]. Cowpea suffers from drought stress due to erratic rainfall due to climate change, resulting in a yield loss of up to 35–69% [57]. To provide a better solution to drought stress in cowpea production in SSA, a comprehensive evaluation and characterization of cowpea genotypes for developing drought-tolerant varieties using physiological, biochemical, and molecular approaches could be exploited. Understanding drought tolerance gene expression will further advance tolerance breeding.

#### 3.2.2. Salinity

Salt stress is one of the most significant abiotic constraints to cowpea crop productivity and severely hampers crop production, especially in arid and semi-arid areas. Cowpea is unfavorably affected by salinity stress at seed germination and seedling stages, and growth and vigor are reduced, which is exacerbated by climate change effects [58]. Salinity stress ultimately reduces the yield (leaves, immature pods, and grain weight and quality) of cowpea, while other vegetative growth traits are also adversely affected [59,60]. In addition, salinity reduces the ability of cowpea crops to take up water and soil-plant nutrients, leading to growth reduction and metabolic changes similar to those caused by low soil moisture stress [61]. Furthermore, it reduces lipid peroxidation and leads to destructive oxidation, which in turn causes damage to the key plant biomolecules [62]. Salt stress is a complex trait, and it is associated with other agronomic and biochemical traits of cowpea. Therefore, for developing salt stress-tolerant cowpea genotypes, integrative (biochemical, molecular, and conventional) breeding approaches are the best solution for tackling the existing problems in SSA.

#### 3.2.3. Heat Stress

Heat stress is a crucial abiotic stress that significantly affects the growth and yield of cowpea [14]. The effects of heat stress on yield and yield-contributing traits include flower and leaf drop, poor pollen fertility and germination, low pod setting, low plant biomass, low harvest index, poor pod filling, and low seed weight and yield [63,64]. Furthermore, physiological and biochemical traits of cowpea negatively affect the photosynthetic apparatus, such as impaired photo-assimilation, inhibited N2 fixation, increased leaf senescence, decreased canopy temperature, and leaf relative water content [64,65,66]. When the night temperature reaches about 16 °C, cowpea flowers abort due to poor pollen development, which causes a 4 to 14% loss in leaves, immature pods and grain yield and quality [18]. In general, using conventional, physiological, phenomic, functional genomic, proteomic, and metabolomic breeding techniques [66,67,68,69,70] and viable approaches to developing heat stress tolerance and to sustain cowpea yield under increasing high-temperature stress, the cowpea germplasm needs to be screened to identify ‘stress adaptive’ traits across various gene pools.

#### 3.2.4. Low Soil Fertility

Phosphorus is essential for cowpea production in many tropical African soils with the inherent low soil availability of phosphorus [71]. Cowpea does not require too much nitrogen fertilizer because it fixes its nitrogen from the air using the nodules in its roots [72]. However, phosphorus is critical to cowpea yield because it stimulates growth, initiates nodule formation, and influences the efficiency of rhizobium-legume symbiosis [73]. Therefore, cowpea requires more phosphorus than nitrogen in the form of single super phosphate [74]. In addition, it is required in large quantities in young cells such as shoot and root tips where metabolism is high and cell division is rapid [75]. It also aids in flower initiation and seed and fruit development [76].

## 4. Breeding Opportunities and Nutritional Profiles of Cowpea Leaves

Cowpea, in Africa and parts of Asia and the American countries, is called ‘the poor man’s meat’ as it is a significant and cheap source of protein, minerals, and vitamins [77] for rural poor people who have limited access to protein from animal sources such as meat and fish [48,49]. It is a nutritious food source, as it is rich in protein and minerals, digestible and non-digestible carbohydrates, and potassium and has a very low lipid and sodium content [77,78].

Fresh cowpea leaves are used as a vegetable, and the haulms, pods, stems, and leaves are used as livestock fodder, providing dietary nutrients for animals and humans [49,77,79,80]. Moreover, all of these components are high in protein, low in fat, and are a vegetable source for human consumption [11,80].

### 4.1. Protein Quality and Quantity

Cowpea is a dry land legume crop consumed as a high-quality plant-based protein source in many parts of the world, particularly in the tropics [6]. In addition, this crop has been promoted as a high-quality protein part of the daily diet of economically depressed people in developing countries to reduce the high prevalence of protein and energy malnutrition [81,82,83,84,85]. Cowpea grain has relatively low-fat and high total protein content [6]. On average, the protein content of cowpea leaves is between 27 to 43% (Table 1), while that of dry grain is 21 to 33% [80], with a high amount of essential amino acids like lysine and tryptophan [86]. Nevertheless, cowpea protein compared to animal protein is deficient in methionine and cysteine [87].

### 4.2. Minerals

Cowpea grains, leaves and immature pods are a source of essential minerals (Ca, Cu, Fe, K, Kg, Mg, Mn, Na, P and Zn, Al) that are required for human health, growth, and development [84,92]. Moreover, Al, B, and Se are present in cowpea leaves and grain [98]. According to previous studies, the amount and availability of minerals in cowpea leaves and immature pods are much higher than in the grain (Table 2). The availability of some minerals like P, K, and Mn in cowpea grain varies widely due to environmental conditions [78]. Macro and micronutrients are essential for the physiological functions of the human body [94].

### 4.3. Vitamins

Cowpea is a source of different vitamins (Table 3). It is rich in vitamin A and C [99,105,106] and has an appreciable amount of B complex vitamins (thiamine, riboflavin, pantothenic acid, pyridoxine, and folic/folate acid) [84]. The vitamin E composition seems to differ significantly from that of most other legume crops, where γ-tocopherol dominates [86]. Vitamin C composition in cowpea leaves is 4- to 38-fold that of the grain [100].

### 4.4. Fatty Acids/Lipids

Compared to chickpea, lentil, green gram, and lupin, cowpea has a low lipid content [91]. The lipid content of cowpea grain and leaves ranges from 0.5% to 3.9% and 1.3 to 4.3%, respectively (Table 4). The lipid profile of cowpea indicates a predominance of triglycerides (41.2% of the total fat), followed by phospholipids (25.1% of total fat), monoglycerides (10.6% of total fat), free fatty acids (7.9% of total fat), diglycerides (7.8% of total fat), sterols (5.5% of total fat), and hydrocarbons plus sterol esters (2.6% of total fat) [86,102]. Palmitic and linoleic acids predominate, followed by oleic acid, stearic acid, and linolenic acid [49]. The main component in the sterols is stigmasterol (42.1 to 43.3%), followed by β-sitosterol (27.6 to 39.5%). In the tocopherol fraction of cowpea seed oil, γ-tocopherol ranged from 44% to 67%, followed by δ- tocopherol (30.3 to 52.8%) [102]. The oil content in cowpea grain is relatively low. Still, it has an extremely high content of biologically active compounds (tocopherols in the oils range from 3838 to 11475 mg/kg, phospholipids 12.2% to 27.4%) [102].

### 4.5. Carbohydrates

Most of cowpea grains, leaves, and immature pods consist of carbohydrates (Table 4). Cowpea leaves and grain are an excellent source of carbohydrates, ranging between 30.39 to 31.11% [62,105] and 50 to 60%, respectively, which makes cowpea a potentially important nutritional component in the human diet [104]. The carbohydrate fractions in cowpea are sucrose (11 to 19 g/kg), glucose (4 to 5 g/kg), fructose (1 to 2 g/kg), galactose (≤15 g/kg), and maltose (≤11 g/kg). Anti-nutrient components of carbohydrates of cowpea are stachyose (17 to 60 g/kg), verbascose (6 to 13 g/kg), and raffinose (5 to 10 g/kg) [86].

### 4.6. Pharmacological Benefits of Cowpea

Globally, meat-based food systems have some drawbacks compared to a vegetable-based food system because it requires more energy for digestion [106]. Protein from cowpea has good nutritional properties and health benefits [106,107]. Cowpea is also a source of many health-promoting components, such as soluble and insoluble dietary fiber, phenolic compounds, minerals, and other functional compounds, including B complex vitamins and tocopherols, anthocyanins, and carotenoids [105,106,107].

Consumption of cowpea exerts protective effects against several chronic diseases [110,111,112]. Among the health benefits of cowpea are low glycemic index carbohydrates [90], positive effects on gastrointestinal disorders [112], cardiovascular diseases, hypercholesterolemia, and obesity [113], as well as anti-diabetic [114], anti-cancer [115], anti-inflammatory [116], anti-hypertensive and hypocholesterolemic [117] properties, and positive effects on insomnia, osteomalacia, osteoporosis, anencephaly, rickets, and cardiac health and metabolic wellbeing [17]. Furthermore, consumption of cowpea protein has been linked to reducing plasma low-density lipoprotein [118].

## 5. Cowpea as a Climate-Resilient Crop

Climate change is arguably the most severe global challenge facing the planet in the 21st Century, as temperatures continue rising, triggering a host of extreme weather events such as heatwaves, droughts, and flooding [119]. Climate change affects the growth of crops through multiple mechanisms, including changing phenology, heat stress, soil fertility, and water stress (frequent and prolonged drought) [120]. Increasing climate variability is, arguably, one of the greatest challenges to food security, particularly through its effects on the livelihoods of low-income people in marginalized communities, which have less capacity for coping, and who depend on highly climate-sensitive crop production activities typical of developing countries [121]. Climate change, along with current agricultural practices, is going to pose a significant challenge for future food and nutrition security, particularly in developing countries [122].

Food security is the main challenge in developing countries, especially in the least developed countries [123]. Orphan crops play an important role in global food and nutrition security for the livelihood of resource-poor farmers [124] and may have the potential to contribute to sustainable food systems under climate change [125]. Orphan crops like cowpea are indigenous and are invariably grown by small and marginal farmers under a subsistence farming system [124].

Cowpea is a future smart food with a high contribution to food and nutritional security and is good for medicinal purposes as well as being an option in extreme environments [88]. Among underutilized pulse crops, cowpea is the most nutritionally dense and climate-resilient, as it is amenable to diverse cropping systems and is locally available for economic growth and social development [122,125]. Cowpea is a multipurpose legume crop. It has high-quality protein and is rich in macro and micro-nutrients for human consumption. It is also rich in protein for livestock fodder and for human consumption. It improves soil fertility by recycling nutrients through nitrogen fixation in association with nodulating bacteria [86] and is weed suppressing [126,127]. This crop is more drought-tolerant than other pulse crops [4].

Cowpea as a vegetable contains important nutrients, including vitamins and minerals, that can improve the nutritional status of individuals and households with proper utilization [103]. The high nutritional value of cowpea leaves makes them ideal for efforts aimed at reducing food and nutrition insecurity. Cowpea can provide considerable amounts of bioavailable nutrients that are useful to alleviate nutrient deficiency among rural and urban populations. Plant-derived minerals and protein are the cheapest alternatives to circumvent the malnutrition that is prevalent in SSA [84]. In general, cowpea can be used as a climate change-resilient legume crop and as a future super crop in marginal environments (Figure 2).

## 6. Cowpea Genetics and Breeding Progress

Cowpea is an annual self-pollinated diploid (2n = 2x = 22), warm-season, multifunctional legume grown for food, fodder, vegetable, and green manure [127] with a 620 Mb genome size [128]. Different breeding methods for cowpea improvement have been widely and successfully used, including pure line selection, mass selection, pedigree, backcross, and single-seed descent [129]. Earliness, growth habit, resistance to biotic stress, drought tolerance, a high and stable yield, and good nutritional profile and quality are the most important traits for cowpea breeding [16,25,129]. For agronomic trait improvements in cowpea, pure line selection, mass selection, pedigree, backcross, and single seed descent have been used, utilizing additive, dominance, additive × additive, additive × dominance, and dominance × dominance effects different traits [130]. Several traits have been improved through conventional and molecular breeding tools to harness cowpea genetic variation for breeding [131].

### 6.1. Conventional Breeding

Leaf shape, leaf size, leaf number per plant, pod number per plant, and pod length are important characteristics that can be used for classifying and distinguishing cowpea varieties. However, few breeding methods have focused on the leaves compared to seeds. Leaf shape can also be used as a morphological or physical characteristic during the selection process if it is closely linked with an agronomic trait of interest. Narrow or hastate leaf shapes characterize most cowpea wild relatives, while cultivated genotypes have the ovate or sub-globose leaf shape. However, a possible adaptive advantage for narrow leaves in wild cowpea has not been investigated. The hastate leaf shape was shown to be dominant to the ovate leaf shape, which could be attributed to direct or indirect selection by breeders.

The genetic control of leaf shape is unknown. Other studies have reported that leaf shape is controlled by the number of cell cycles that occur during leaf development [132]. The leaf shape is influenced by environmental factors, including light intensity, temperature, and humidity [133]. The effect of genetic control and environmental factors on the development and the nature of leaf shape is unknown.

Understanding the variation among cowpea genotypes for leaf shape and size could potentially assist breeders in improving the crop for drought tolerance and yield. Previous studies [134] have shown that drought tolerance is associated with small leaf size, while other studies have reported that leaf shape and size are associated with photosynthesis rate and capacity. It has also been reported that leaf shape is not affected by geographic region [135], while other studies [136] have reported the opposite.

Several studies have investigated the variability in the nutritional value of cowpea leaves [59,60] in single experimental trials. Cowpea leaf protein values evaluated in different locations ranged from 25.1 to 43.1% [96,97]. This large variation could potentially be due to genetic and environmental effects. The iron concentration and β-carotene content in the same 561 samples ranged from 140.5 to 3994.7 µg/g and 4.1 to 30.5 mg/100 g, respectively. Contents of up to 18.7 mg of Fe, 0.547 mg of Zn, and 4.45 mg of beta carotene per 100 g of the edible portion of freeze-dried raw cowpea in three districts, Kongwa, Singida and Arumeru, in Tanzania have been reported [136]. In South Africa, mineral concentrations in cowpea leaves have been reported to be 142 to 626 mg·kg^−1^ for Fe, 49 to 104 mg/kg^−1^ Zn, 196 to 394 mg/kg^−1^ Mn, 8.6 to 19.7 mg/kg^−1^ Cu and 42 to 55 mg/kg^−1^ B [96]. These studies indicated sufficient variability among the tested cowpea genotypes for breeding for improved nutritional value. The above findings also suggested the need to comprehensively investigate the effect of genotype × environment (i.e., environments and seasons) interactions on cowpea genotypes for nutritional value.

Several studies have focused on investigating genotype × environment (G × E) interactions and the change in the ranking of cowpea genotypes for seed grain in Egypt [100], Brazil [133], Ethiopia [134], Zimbabwe [135], and South Africa [136,137,138]. Fewer studies [97] have focused on G × E for cowpea yield and nutritional values. Another study [97] investigated the variability of 25 genotypes for nutritional value in cowpea leaves in one location for two seasons. Thus, only the genotype × season effect was evaluated [138].

### 6.2. Molecular Breeding

Molecular tools and genomic resources have been developed for cowpea crop improvement [139,140,141,142,143,144,145]. These integrated genomic resources include a 1536 single nucleotide polymorphism (SNP) genotyping platform, and an EST-derived SNP consensus genetic map has been developed. Using the same SNP markers, a cowpea physical map has been partially anchored to the cowpea consensus genetic map. About 500 diverse cowpea accessions have been SNP-genotyped, and the initial cowpea genome has been assembled. These resources will enable the dissection of underlying genetic components of target agronomic traits using Quantitative Trait Locus (QTL) analysis and association mapping.

Numerous biparental populations have been used to map major QTLs for various traits [140,141,142] and to develop consensus genetic maps of cowpea [144,145]. For example, one study [146,147] focused on using the SNP markers to map the hastate versus ovate leaf shape trait in a biparental recombinant inbred line (RIL) population. In addition, new populations have been developed for higher-resolution mapping, including eight parents and 305 lines [140]. Recently, a reference genome sequence of cowpea [144] and genome assemblies of six additional diverse accessions [128] have been produced. The identified and confirmed QTLs would facilitate cultivar improvement using marker-assisted breeding.

#### 6.2.1. Marker–Trait Associations

Marker–trait association analysis has become a valuable tool in functional plant genomics and high-resolution mapping of QTL [148,149]. Several linkage maps have been used to identify QTLs for desirable traits in cowpea breeding (Table 5). In cowpea, understanding flowering time is a key player in plant adaptation and is an important phenological trait for breeding because agronomic traits such as plant growth, plant height, pod number, pod length, and nutritional quality traits depend on the time of flowering [150]. Previous studies have been focused on identifying QTL using SNP and simple sequence repeat (SSR) markers in RILs. SNP and SSR markers were utilized in another RIL population of ZN016 × ZJ282 to identify QTLs for days to flowering, nodes to first flower, leaf senescence, and pod number per plant [151].

#### 6.2.2. Transcriptomics

Transcriptomics (microarrays, RNA-seq) is the complete set of transcripts in a cell and their quantity for a specific developmental stage or physiological condition. Transcriptome analyses may help to understand how the transcriptome changes contribute to various cellular processes, gene discoveries, and putative gene functions [157,158,159,160,161]. In cowpea, some researchers have been using the transcriptomics approach. In IT97K-499-35, a total of 27cDNA libraries were generated from the major vegetative tissues (roots, stem, and flower of five-week-old plants) (http://vugea.noble.org accessed on 19 April 2022). Using the roots, stem, and leaves of cowpea seedlings of 32 cowpea accessions with the Illumina sequencing techniques, 54 million high-quality cDNA sequences were identified and assembled into 47,899 unigenes containing 5560 genic SSR markers [162,163]. These results showed that genic-SSRs are valuable genetic resources for use in various breeding strategies to increase the efficiency of cowpea improvement where molecular markers and genomic selection resources are limited.

Early developing seed tissues of two cowpea genotypes (IT86D-1010 and IT97K-499-35) using the Illumina HiSeq 2500 system generated 125 to 265 million reads, which were assembled into 35,000 to 74,000 transcript contigs [164] and were used to develop transcriptomic resources to characterize expressed genes in leaves and notably floral tissues undergoing male and female gametogenic development and early seed initiation. The majority of the transcript contigs across all the tissues in both genotypes could be mapped to the cowpea v1.0 reference genome. This is important for future exploration of cis-regulatory regions associated with tissue-specific gene expression. Distinct changes in global gene expression profiles that occur in host roots following successful and unsuccessful attempted parasitism by Striga may help to understand the resistance mechanism. The induction of specific defense-related genes and pathways defines the components of a unique resistance mechanism [165].

### 6.3. Genetic Resource Management

Crop genetic resources are an essential component of agrobiodiversity. The genetic materials of crops have value as a resource for present and future material developments for sustainable uses [166]. Cowpea germplasm resources are also strategic resources that are essential to national and global agricultural security through sustaining the desirable traits on the target crops for crop breeding, research, and conservation management and the long-term resiliency of food security. The International Institute of Tropical Agricultural (IITA) maintains the world’s largest collection of cowpea germplasm of over 15,003 accessions from 90 countries in its gene bank [167]. More than 80% of accessions were characterized for 28 agro-botanical descriptors of cowpea [167,168]. In addition to IITA, the United States Department of Agriculture-genetic Resources Information Network (USDA-GRIN) at Griffin, the USA and the University of California, Riverside, USA, are also conserving cowpea collection of about 7737 and 6000 accessions, respectively [169,170]. At this time, there is a larger war on seed ownership in the African content and globally [171]. A comprehensive collection, conservation, and cowpea characterization for each growing region will be crucial for future development of new varieties and identification of desirable genes for specific traits. The existing and future collections are sources of genes that are needed for enhancing the productivity of new, improved varieties in short, medium, and long-term strategies for climate change resilience.

## 7. Breeding Strategies and Research Perspectives for Cowpea

The application of modern genomic selection and agronomic tools to improve indigenous/orphan crops can provide enormous and novel opportunities for ensuring global food and nutritional security [172]. Developing desirable cowpea varieties for sustainable uses in tropical and sub-tropical regions would employ a wide range of conventional and contemporary breeding strategies (Figure 3). Implementing a structural breeding program that takes advantage of additional modern crop improvement tools such as genomic selection, speed breeding, genomic editing, mutation breeding, transgenic approaches, high throughput phenotyping, and breeding digitization would allow rapid improvements to orphan crops [173].

### 7.1. Genomic Selection

Genomic selection (GS) is a promising approach for exploiting DNA markers to design novel breeding programs and to develop new marker-based models for genetic evaluation and unlocking of traits [174]. In cowpea breeding as a vegetable, GS has opened new avenues to implement simultaneous selection for several traits [175], and provides opportunities to increase the genetic improvement of complex traits per unit of time and cost [175].

In this crop, genomic strategies have significantly accelerated the development of climate-smart genetic resources or elite vegetable varieties [131,132,176], through identifying genetic diversity and favorable variation required for climate resilience and food and nutritional security, identifying traits and genes for tolerance to new and complicated stresses induced by climate change and taking integrated genomics tools and approaches to manage combining tolerance to abiotic and biotic stresses as well as improved nutritional profiles and quality of grain, leaves and the immature pods of cowpea [177]. Increasing genetic gains and breeding efficiency through a rapid-cycle genome-wide selection could be a future cost-effective and precise method for orphan crop improvement for human food and animal feed and rehabilitation of poor soils of overexploited cultivated land [178].

### 7.2. Speed Breeding

The current world population is 7.8 billion and is projected to reach 9.7 billion by 2050 [179]. Climate fluctuations involving rising temperatures, more frequent floods, and frequent and prolonged drought episodes are predicted and could lead to frequent pest outbreaks and new disease pandemics with high severity. In addition, agricultural production should be increased by 60–110% to meet the global requirement of the projected population by 2050 [180]. Around 2 billion people are facing extreme micronutrient deficiencies, and more than 815 million are suffering from chronic hunger [181]. Therefore, tackling existing and upcoming challenges with a fast and effective response requires accelerating plant breeding as one of the best fitting approaches [182] and the key technologies that would revolutionize the breeding of orphan crops [183]. Speed breeding protocols could be applied to shorten breeding cycles and accelerate the improvement activities of orphan crops [183]. Essentially, speed breeding (SB), commonly known as fast-tracked breeding, is an additional tool available to speed up plant breeding [184]. SB is the deliberate manipulation of various growing conditions and has been used on various crops to rapidly develop iso-lines after initial crosses of target parents with complementary traits. The techniques depend on the manipulation of photoperiod, light intensity, temperature, soil moisture, soil nutrition, and high-density planting for developing an elite variety for immediate use [185]. SB, therefore, could be very promising when implemented in the right way to enhance underutilized crops [183].

### 7.3. Mutation Breeding

Climate change is rapidly changing how we live, what we eat, and how we eat and produce the underutilized crops we breed and the target traits [173]. For more than 70 years, mutation breeding has been used for crop improvement for different traits [186]. Mutation breeding has been used for improving both the oligogenic and polygenic traits of many crops. It has been employed to enhance morphological and physiological traits, biotic and abiotic stress resistance, yielding ability and nutritional quality, growth habits, and preferred end-user characteristics in crops for better trait selection and in order to contribute to global food security [186,187]. The application of numerous induced mutations was used to correct one or a few negative characters and to get new gene combinations which are desirable without changing the plant’s total genetic makeup [187]. In creating variability through mutagenesis, either physical (fast neutron, UV, X-ray, and gamma radiation) or chemical (N-methyl-N-nitrosourea (MNU), sodium azide, hydrogen fluoride (HF), methyl methanesulfonate (MMS), or ethyl methanesulfonate (EMS) have been widely used for new material development [188] and are important for developing improved varieties in the field of functional genomics [141,144]. In addition, Agrobacterium and transposon-based chromosol integration for complex traits like yield are important [189]. Globally, over 3000 new varieties have been released using mutation breeding, including some neglected crops in Africa [129]. Mutation breeding helps to enhance the tolerance of crops to diverse climatic conditions such as high temperatures, drought, and the occurrence of insect pests and diseases [141,145].

### 7.4. Genome Editing

Genome or gene editing provides precise, heritable genome mutagenesis without permanent transgenesis, and has been widely demonstrated in crops using a conventional method that is applied to alter the genotype and phenotype of crops and comprises the use of both natural and artificial crosses and induced mutations [190] and modern genome editing [191] approaches for crop improvement. Currently, genome editing (GE) could be used to rapidly modify undesirable traits in neglected crops to accelerate the process of domestication and for sustaining food and nutritional security [192].

Compared to the well-studied crops, minor crops have limiting factors, including a lack of annotated genomes, sub-optimal tissue culture regeneration protocols and a lack of genetic transformation methods [183]. Therefore, to fill these existing GE knowledge gaps, a knowledge model or the study of other crops or more closely related species can be used to translate genetic knowledge from one crop to another [193]. Thus, as more genome sequences become available, it becomes easier to understand and use the information for crops currently being domesticated and to identify orthologs of known domestication genes [192]. In recent years, rapid modern GE technologies, such as zinc finger nuclease (ZFNs), transcription activator-like nucleases (TALENs), and cluster regularly interspaced short palindromic repeats/CRISPR associated protein (CRISPR/Cas), meganucleases and targeted regularly double-strand breaks (DSBs) have been developed [193], which demonstrates the broad applications used in the discovery of new genes or undesirable gene functions [192].

## 8. Policy and Regulatory Framework

A country’s national policies have a direct impact on the adoption and expansion of cowpea as a climate-smart crop, and without support from local leaders and policymakers, use of this crop as a food security crop is limited. Many SSA countries are still lacking in the capacity and essential procedures required to establish adequate policies, regulations, implementation, and monitoring frameworks regarding seed systems (including the whole value chain) [194,195] and the development and production of multi-nutrient purposes of cowpea. In SSA countries, the major bottleneck to the adoption of the genetic modified organism (GMOs) and deployment of Bt-expressing cowpea in Africa is the lack of political will and of external influence. Leaders and policymakers should be able to make informed decisions and publicly defend such decisions on GM technology. Therefore, the involvement of scientists, industrial research councils, and policymakers should work together to create awareness and advocate policy changes to permit GM technology.

## 9. Conclusions

Cowpea is a future smart food crop with excellent nutritional and nutraceutical properties. It has several agronomic, environmental, and economic advantages in underdeveloped and developing countries, contributing to food and nutritional security. Cowpea is truly a multifunctional climate-resilient crop for promoting global food security and maintaining good health to reduce non-communicable human diseases. In addition, cowpea leaves and immature pods can be used as food by millions of people in the developing world and can be used as a food ingredient in the food industry.

However, biotic (viral, fungal, and bacterial diseases, nematodes, aphids, parasitic weeds, and several insect pests) and abiotic stress (such as phosphorus stress in the soil; heat and drought stress), cultural beliefs, low yield, limited attention from researchers, limited developments, as well as policymakers and limited modern breeding technology resources for cowpea breeding as a vegetable crop, are major challenges for this neglected crop.

Nutritionally rich cowpea vegetable cultivars can be developed using existing genetic resources for target traits, and with the integration of chance breeding (mutation, hybridization, backcrossing, pedigree, and recurrent selection), modern breeding approaches (space breeding, speed breeding, genomic selection, and genome editing), technological innovation (mutagenesis breeding), transgenic approaches, demand-led breeding and multi-omics approaches would be the best solution for tackling the existing production problem in SSA.

With the use of all possible breeding methods and approaches, future cowpea crop improvements should focus on breeding for high yielding cultivars with good nutritional quality traits and resistance to biotic stresses and tolerance to abiotic stresses. High yielding cultivars with a good nutritional value, as well as resistance and tolerance to diseases and pests, could potentially be used as parents for future crop improvement, which will result in the development of superior varieties. Successful breeding for superior cowpea genotypes is expected to contribute to food security and to help combat malnutrition in Africa and beyond.

## Figures and Tables

**Figure 1 plants-11-01583-f001:**
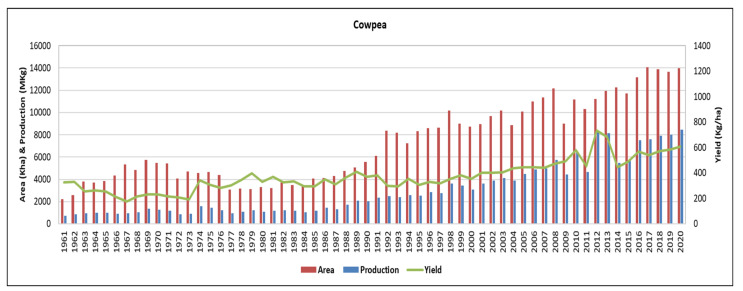
Trends in area, yields and production of cowpea in SSA. The source adapted from [19].

**Figure 2 plants-11-01583-f002:**
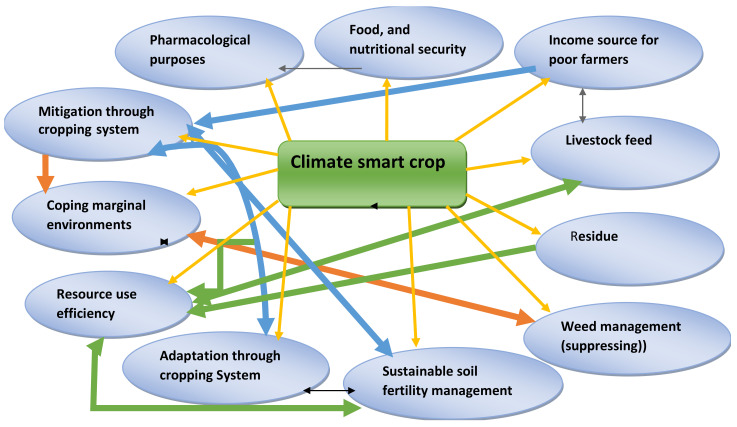
Cowpea as a climate-smart crop for SSA.

**Figure 3 plants-11-01583-f003:**
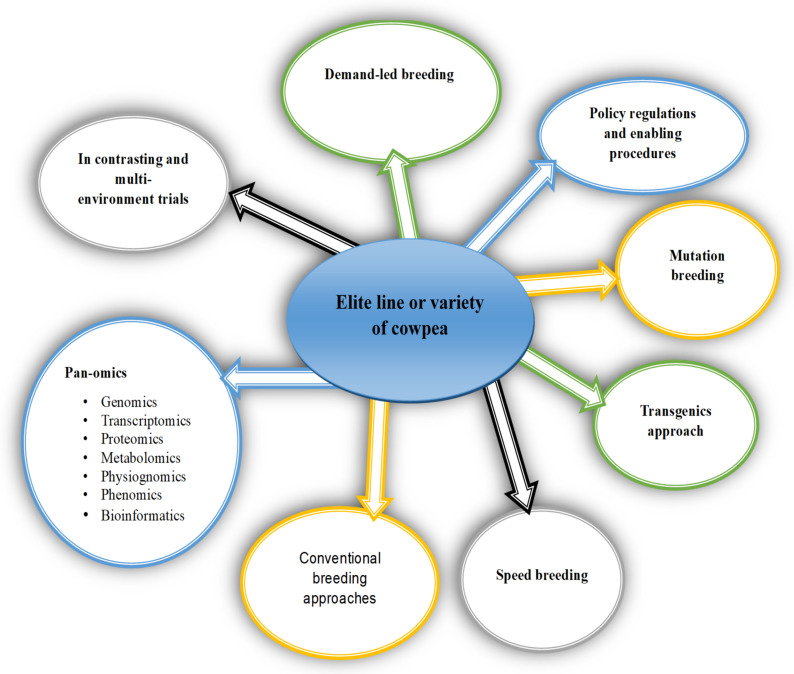
Breeding strategies and research perspectives of cowpea for food and nutritional security.

**Table 1 plants-11-01583-t001:** Amino acid composition of cowpea grain and leaves.

Amino Acid	Leaves (g/100 g Protein)		Grain (g/100 g Protein)	
	Mean Range	References	Mean Range	References
Aspartic acid	10.8–26.7	[86,88]	6.0–13	[86,89,90]
Arginine	7.4–17.3	[88,91]	5.0–10.8	[86,90]
Alanine	4.2–9.8	[91,92,93]	3.4–5.1	[10]
Methionine	1.0–4.5	[91,94]	0.9–3.5	[91,94]
Glutamic acid	17.2–45.3	[91,93]	8.5–19	[91,95]
Glycine	3.8–12.6	[91,93]	3.1–4.8	[94,96]
Cysteine	0.5–2.9	[86,93]	0.3–2.4	[91,96]
Histidine	1.8–8.6	[91,94]	2.0–4.41	[94,97]
Isoleucine	4.1–11.1	[84,91]	2.8–5.4	[94,97]
Leucine	7.4–19.6	[91,94]	5.7–11.3	[94,97]
Lysine	3.0–16.3	[91,94]	3.5–8.0	[5]
Phenylalanine	4.6–14.4	[91,94]	4.4–9.9	[94,95]
Proline	4.0–15.9	[91,93]	3.1–8.9	[91,95]
Serine	3.0–11.6	[91,93]	3.8–5.8	[94,95]
Threonine	3.2–10.8	[84,91]	3.0–5.9	[94,97]
Tryptophan	1.3–4.1	[91,93]	0.9–1.5	[94,95]
Tyrosine	3.0–9.3	[91,93]	2.6–4.5	[5]
Valine	5.0–12.8	[91,93]	3.4–6.2	[91,94]

**Table 2 plants-11-01583-t002:** Mineral composition of cowpea grain, immature pods, and leaves.

	Leaves		Immature Pod		Grain	
Minerals	Mean Range	References	Mean Range	References	Mean Range	References
Macro-minerals (mg/100 g dry matter)						
Calcium	15.2–46.20	[15]	223.67–867.77	[16]	0.07–2.7	[89,94]
Phosphorus	2.3–6.10	[15]	383.43–537.53	[16]	2.1–592.4	[89,94]
Potassium	9.30–35.60	[15]	170.74–240.78	[10,85]	9.57–1445.2	[89,95]
Magnesium	4.3–8.4	[15]	297.97–426.20	[16]	1.3–227.4	[89,99,100,101]
Sulfur	153.3–200.0	[24]			120.0–147.3	[42]
Micro-minerals (mg/100 g dry matter)						
Copper	0.15–2.2	[101,102,103]	0.48–0.95	[16]	0.5–2.2	[95,101]
Iron	26.76–182.33	[25]	6.01–9.78	[16]	3.4–10.6	[89,104]
Manganese	10.57–204	[101,103]	2.11–4.77	[16]	1.38–4.3	[89,101]
Sodium	11.59–43.95	[25]	13.70–32.93	[16]	8.4–79.81	[89,95]
Zinc	2.78–22.3	[101,103]	1.42–5.63	[10,85]	2.4–5.11	[89,94]
Aluminum			1.84–7.86	[16]		
Boron	3.14–5.01	[27]	2.13–4.03	[16]	1.47–2.14	[27]
Selenium			2.5–3.4	[23]		

**Table 3 plants-11-01583-t003:** List of vitamins in cowpea grain.

Vitamins	Mean Range/Mean (%)	References
Vitamin A	0.00–0.07	[5]
Vitamin B1	0.2–1.7	[5]
Vitamin B2	0.1–76	[91,107]
Vitamin B3	0.7–4.0	[5]
Vitamin B5	1.7–2.2	[5]
Vitamin B6	0.2–0.41	[5]
Vitamin B7	0.02–0.03	[5]
Vitamin B9	0.1–0.4	[5]
Vitamin B12	0 or trace	[5]
Vitamin C	1.5–1.69	[28]
Vitamin D	0.00	[28]
Vitamin E	0.07–20	[5]

**Table 4 plants-11-01583-t004:** Proximate and fiber composition of leaves and grain.

Nutrient	Leaves		Grain	
	Mean Range (%)	References	Mean Range (%)	References
Moisture	8–9	[30]	11.81–13.24	[31]
Ash	8.1–14.4	[10]	3.1–5.8	[10]
Crude protein	27–43	[32]	21–33	[32]
Crude lipid	1.3–4.1	[10]	0.5–3.9	[10]
Crude fiber	10.09–35.9	[94,108,109]	18–32	[11]
Carbohydrate	59.7–65.2	[33]	50–60	[34]

**Table 5 plants-11-01583-t005:** Breeding achievements of cowpea using marker–trait association.

Traits	Population	Type	Marker Type	QTLs	References
Cowpea golden mosaic virus	IT97 K-499-35 × Canapu T16	F2	AFLP	3	[147]
Fusarium wilt resistance	CB27 × 24–125B-1	RIL	SNP	1	[148]
Days to flowering	524B × 219-01	RIL	SSR	3	[152]
Pod length	(JP81610 × TVNU457) × JP81610	BC1F1	SSR	9	[153]
Pod tenderness	(JP81610 × JP89083) × JP81610	BC1F1	SSR	3	[154]
Foliar thrips	CB46 × IT93 K-503-1 and CB27 × IT82E-18	RILs	SNP	3	[155]
Cowpea bacterial blight resistance	Danlla × TVu7778	RIL	SNP	3	[155]
Charcoal rot resistance	IT93 K-503-1 × CB46	RIL	SNP	9	[150]
Striga gesnerioides	TVx 3236 × IT82D-849	F2	AFLP	3	[151]
Hastate leaf shape	Sanzi × Vita 7	RIL	SNP	1	[148]
Pod fiber layer thickness	524B × 219-01	RIL	SSR	4	[156]
Pod number per plant	ZN016 × ZJ282	RIL	SSR	3	[146]
Pod tenderness	(JP81610 × JP89083) × JP81610	BC1F1	SSR	3	[154]
Nodes to the first flower	ZN016 × ZJ282	RIL	SNP	4	[146]
Days to first flowering	ZN016 × ZJ282	RIL	SNP	3	[146]
Days to maturity	IT93K503-1 × CB46	RIL	AFLP	2	[155]

## Data Availability

Not applicable.

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
