# Peer review of "Breeding of Vegetable Cowpea for Nutrition and Climate Resilience in Sub-Saharan Africa: Progress, Opportunities, and Challenges"

_plants, 2022, doi:10.3390/plants11121583_

Round 1

Reviewer 1 Report

Lines 18-20 ... contribute to resilience is increasing, and humanity is facing food and nutritional scarcity to marginal cropping systems and improve agricultural sustainability under climate change’. These statements are not clear. Need to be re-written.

Lines 39-40... ’The crop can be grown under harsh conditions such as drought and sandy soils’. Is there any reference to support this?

Lines... 76-80. Actually, the authors didn’t provide information about growing area and production share of cowpea in SSA area at the moment. Namely, is there any other crop in SSA area which dominate over  cowpea by area and production at present? If yes, what is possibilities of cowpea to replace that crop in near future? What is advantages/diadvantages of cowpea over most wide-spread crops in SSA in sense of climate change? So,  some kind of comparative analyses between cowpea and other major crops that could be replaced by cowpea  in future is needed.

Regarding constrains to cowpea production. Bacterial and viral diseases, root-knot nematodes can reduce yield up to 100%. Can you provide possible solutions for these problems, as you stated some for parasitic weeds (lines 132-134) and insect pests (lines 152-154)?

Table 4 title: what does it mean Proximate...?

Line 232... santos and boiteux, 2013 replace with number

Previous studies have shown that leaf size and shape are associated with drought tolerance and photosynthesis rate and capacity (lines 291-293). Is there any link betwen leaf size and shape with nutritional properties? Is there any other traits except leaf shape and size suitable for breeding/that showed high variability? Finally, any plant breeding programme starts with evaluation of diverse germplasm. Are there collections of cowpea genotypes in public gene banks worldwide (it should be mentioned in the text) or there are just (private) breeding collections within breeding companies?

Lines 381-382...To be re-written. Not cleart, some word are in duplicate...

Is there any example of mutation breeding in cowpea to be cited?

Line 439... what is chance breeding?

Supported polices are mentioned several times in the text, but there is no any explanation/further details about this.

English need to be edited for missing/doubling words, unclear statements, signs of interpunction etc.

Author Response

Dear reveiwer,

Thank you so much for the comment. As per the comments of you, the academic editor, and the other reviewers, the manuscripts were highly improved. 

 I have attached the responses.

Thank you so much for your contribution.

Tesfaye

Reviewer 2 Report

Comments for authors

The review “Nutritional contribution of cowpea to sub-Saharan countries as a climate resilience crop” aims to highlight cowpea crop value as a vegetable resource that could play an important role in global food and nutritional security under climate change conditions and in malnutrition prone regions of sub-Saharan Africa (SSA) countries

Here are some considerations:

Line 40 - Thus, its ability of to tolerate climate changes makes it an important legume crop for food and nutritional security in tropical and sub-tropical regions of the world, including sub-Saharan Africa (SSA).

No bibliographic reference to cowpea as resilient change climate crop

Line 81 - 3. Constraints to cowpea production

Section 3 lists the abiotic and abiotic stresses. On the former, apart from a simple list, no data and bibliography are given; biotic stresses, on the contrary, are in more detail described in various subsections. Some more detail on abiotic stresses would be appropriate.

Line 172 - On average the protein content of the leaves is between 27 to 43% (Table 1), while that of dry grain is 21 to 33%.

Table 1 actually shows amino acid composition of cowpea grain and leaves

Line 223 -Pharmacological benefits of cowpea

It is not made explicit how cowpea is able to reduce both communicable and non-communicable human diseases, as reported in the abstract (line 27) and conclusion (line 430)

Line 273 – 6. Breeding approaches for cowpea as a vegetable - In section 6.1 the authors are advised to review this part. There is much talk about leaf shape and that most wild relatives of cowpea are characterised by narrow leaf shapes while cultivated genotypes have an ovate shape.

The authors speculate that leaf shape can be used in breeding programmes if it is associated with certain valuable agronomic traits. This has not been studied. Furthermore lines 296-297 I would like to ask the authors with reference to the protein content of the leaves whether two literature references [67][68] are sufficient to state a range of variability between 25.1 to 43.1%. Do these values seem very high?

In sections 7.1, 7.2, 7.3 and 7.4, the authors report on various genetic improvement methods of potential use for cowpea with numerous bibliographical references. I do not see the relevance of this part for the reader of Plants. Furthermore lines 361-367 what is reported is not strictly necessary to the specific paragraph

The conclusions give no concrete indication of the research needed to improve agronomically and nutritionally cowpea crop. All known breeding approaches are suggested for developing new varieties.

Kind regards

Author Response

Dear reviewer,

Thank you so much for the comment. As per the comments of you, the academic editor, and the other reviewers, the manuscripts were highly improved. 

 I have attached the responses.

Thank you so much for your contribution.

Tesfaye

Reviewer 3 Report

The review article entitled, "Nutritional contribution of cowpea to sub-Saharan countries as a climate resilience crop" is a well written and interesting one. However, in my view there should be further improvement as follows:

  1. At the end of the abstract, the authors should draw a conclusive part of the entire review with some future questions. I will make the readers more comfortable about the article.
  2. Under the Introduction part, authors must highlight about the type of cowpea. Being a leguminous crop with nitrogen fixation capability, quick ground coverage, it can do miracle under conservation agriculture systems in dry as well as rainfed areas in addition to the nutritional benefits. Authors should focus some other leguminous crops as well in support of the cowpea. Rest is okey in the introduction part. I would suggest to add my concern in one paragraph after line no. 54. Also use the following two references in the introduction part: a. Pramanick, B., Brahmachari, K., Ghosh, D. and Bera, P.S. 2018. Influence of foliar application seaweed (Kappaphycus and Gracilaria) saps in rice (Oryza sativa)–potato (Solanum tuberosum)–blackgram (Vigna mungo) sequence. Indian Journal of Agronomy 63(1): 7–12; b. Pramanick, B., Bera, P.S., Kundu, C.K., Badopadhyay, P. and Brahmachari, K. 2015. Effect of different herbicides used in transplanted rice on weed management in rice–lathyrus cropping system. Journal of Crop and Weed 10(2): 433–436.
  3. Under the constrains, authors mainly focused on disease-pest related problems. however, there are many problems related to crop management, viz. growing in marginal lands with poor management even low fertilization, weed related problems etc. Authors must focus on these aspects as well.
  4. The references cited in 90 and 91 are old enough. Authors are advised to replace those with the following relevant ones: https://doi.org/10.3390/agronomy11112190 and https://doi.org/10.3390/agronomy11081622
  5. I would suggest the authors to include another short paragraph in the conclusion stating the overall insights and future strategies to make this crop more suitable under changing climatic scenario.
  6. Authors are encouraged to create more figures particularly in the climate-smart section (section 5) of this review article.   

Author Response

(The authors gave the same response as above.)

Round 2

Reviewer 2 Report

Dear authors, thank you very much for accepting my suggestions.  I suggest the publication of this manuscript as it is.

Best regards.

Author Response

Dear Reviewer,

Thank you so much for your valuable comments and suggestions on our manuscripts.  I, on behalf of the authors, also thank you for the acceptance of our corrections.

Regards,

TW Mekonnen